# Osteoporosis Associated with Breast Cancer Treatments Based on Types of Hormonal Therapy: A Cross-Sectional Study Using Korean National Sample Data

**DOI:** 10.3390/medicina59091505

**Published:** 2023-08-22

**Authors:** Yen Min Wang, Yu-Cheol Lim, Deok-Sang Hwang, Yoon Jae Lee, In-Hyuk Ha, Ye-Seul Lee

**Affiliations:** 1Jaseng Hospital of Korean Medicine, 536 Gangnam-daero, Gangnam-gu, Seoul 06110, Republic of Korea; medesu@naver.com; 2Jaseng Spine and Joint Research Institute, Jaseng Medical Foundation, 2F, 540 Gangnam-daero, Gangnam-gu, Seoul 06110, Republic of Korea; hmh6692@gmail.com (Y.-C.L.); goodsmile8119@gmail.com (Y.J.L.); 3Department of OB & GY in Korean Medicine, College of Korean Medicine, Kyung Hee University, Seoul 02453, Republic of Korea; soulhus@gmail.com

**Keywords:** aromatase inhibitors, breast cancer, osteoporosis, selective estrogen receptor modulator, antineoplastic agents

## Abstract

*Background and Objectives*: This study aimed to investigate osteoporosis-related treatments and the overall anticancer drug treatment tendencies, with a focus on selective estrogen receptor modulators (SERMs) and aromatase inhibitors (AIs), in Korean patients with breast cancer from 2010 to 2019. *Materials and Methods*: Data were obtained from the Health Insurance Review and Assessment Service. Patients with breast cancer (International Classification of Diseases, 10th Revision code: C50) as a principal diagnosis at least once from 2010 to 2019 were included. Those with osteoporosis (M80, M81, or M82) as a principal or sub-diagnosis or those who received osteoporosis treatment at least once were categorized as the osteoporosis-related treatment group, and others as the non-osteoporosis-related treatment group. The trends of drug prescriptions and treatment costs in patient groups were evaluated using descriptive statistics. *Results*: Among all included patients, those aged 45–54 years (40.20%) without osteoporosis treatment and those aged 55–64 years (34.11%) with osteoporosis treatment were the most common. SERM was the most commonly prescribed anticancer drug (29.20%) in the entire patient group, followed by AIs (20.83%). Patients without osteoporosis treatment had the highest prescription rate of SERM (31.48%), and those with osteoporosis treatment had a higher prescription rate of AIs (34.28%). Additionally, SERM and AIs were prescribed most frequently before and after the age of 55 years, respectively, regardless of the presence of treatment. *Conclusions*: This study found that osteoporosis-related treatment and patient age were associated with anticancer drug prescriptions. The present findings would help clinicians and researchers in the clinical diagnosis and treatment of breast cancer.

## 1. Introduction

Breast cancer morbidity is on the rise, and it accounts for 25% of the new cancer incidence in women [1,2]. In 2018, the prevalence of breast cancer was estimated to be 46.3 per 100,000 women worldwide [2,3]. In Korea, breast cancer has the highest incidence among cancers in women and is continuously increasing [2], although the mortality rate is relatively low [4]. Additionally, due to its early onset [5] and long survival period, the burden of medical expenses has been increased [6]. A previous study explored the trend of breast cancer-related costs and showed that total socioeconomic and direct non-medical costs in Korea increased by approximately 40.7% and 49.6%, respectively, from 2007 to 2010 [7].

Patients receive targeted therapy, hormone therapy, and radiotherapy and undergo surgery depending on the affected area or progression stage [8,9]. Tamoxifen, a representative prescription for endocrine therapy, has been widely used for decades [10], and aromatase inhibitors (AIs) have become the primary treatment for postmenopausal women [11]. The previous literature indicated the ineffectiveness of AIs before menopause despite their overall efficacy [10,12]. However, recent studies suggest that they are more effective than tamoxifen, even before menopause, when accompanied by ovarian suppression therapy. These findings raise expectations that AIs can generally be a better alternative to tamoxifen [12]. Furthermore, human epidermal growth factor receptor type 2-positive patients, accounting for 25% of patients with breast cancer [13], have been treated with targeted therapies, including trastuzumab (Herceptin), pertuzumab [14], and trastuzumab emtansine [15].

Coping with side effects is an inevitable challenge in managing the health-related quality of life of patients with breast cancer. Common chemotherapy-related complications are nausea, vomiting [16,17], fatigue, muscle pain, diarrhea, and peripheral neuropathy [18]. Another major complication is osteoporosis, affecting up to 80% of breast cancer survivors [6,19]. While osteoporosis is a common risk for postmenopausal women with or without breast cancer, its occurrence in breast cancer patients is primarily attributed to AIs since they deplete residual estrogen and are associated with rapid bone loss and increased risk of fractures [20,21]. Notably, AI-associated bone loss (AIBL) occurs at a rate at least twice as high as bone loss seen in healthy postmenopausal women [20]. To compensate for this side effect, when AIs are prescribed for a long time, adjuvant treatments, such as denosumab, which prevents bone resorption, are commonly administered [22,23].

Given the high survival rate and high prevalence of AIBL among breast cancer patients, effectively managing osteoporosis becomes an important concern. The existing literature recommends the use of osteoporosis medications for both therapeutic and preventive purposes [24,25]. However, only one study has been published 8 years ago on the prescription trend of selective estrogen receptor modulators (SERMs) and AIs, in which a shift from tamoxifen to AI was detected [26]. An insufficient number of studies on osteoporosis-related treatments in breast cancer patients and lack of recent updates deter clinicians and researchers from understanding the latest epidemiological evidence on its prevalence and relation to anticancer treatments.

This study aimed to examine the basic characteristics of patients with breast cancer who underwent osteoporosis-related treatments, including those for preventive purposes, and the treatment tendencies of overall drugs and anticancer drugs, especially SERMs and AIs, from 2010 to 2019. Using national health insurance claims data from the Health Insurance Review and Assessment Service (HIRA), we investigated prescription status of osteoporosis-related treatments based on osteoporosis diagnosis and age. Additionally, we examined the overall treatment cost. This study could provide meaningful fundamental data for establishing better treatment plans for patients undergoing breast cancer and osteoporosis-related treatments.

## 2. Materials and Methods

### 2.1. Data Source

This study used data from the National Patient Sample (NPS), which were obtained from the HIRA from January 2010 to December 2019. HIRA data comprise claims data submitted by medical service providers seeking reimbursement their services from the National Health Insurance (NHI). The HIRA-NPS data are sampled annually by stratifying 2% of the randomly extracted sample (approximately 1 million people) from the entire Korean population (registrants of NHI or Medical Aid) by sex and age. These secondary data are statistically sampled after removing personally identifiable information and corporate data from the raw data, and they consist of medical treatment details claimed within a 1-year period starting from the commencement date of care in that year. They provide detailed and diverse information, such as treatment details, diagnosis, costs paid by insurers, patient copayment, patient demographic characteristics, and information on medical institutions [27].

### 2.2. Study Design and Population

This study included patients with breast cancer (International Classification of Diseases, 10th Revision, ICD-10 code: C50) as a principal diagnosis at least once during the 10 years from 2010 to 2019. Among these patients, those who had either a principal or sub-diagnosis of osteoporosis (ICD-10 code: M80 (osteoporosis with pathological fracture), M81 (age-related osteoporosis without current pathological fracture), or M82 (osteoporosis in diseases classified elsewhere)) or received osteoporosis treatment at least once were classified as the osteoporosis-related treatment group (OSP patient group), and the others were classified into the osteoporosis non-treatment group (non-OSP patient group). Osteoporosis treatment was defined as having prescribed medication classified by the following anatomical therapeutic chemical (ATC) codes: M05BA, M05BB, M05BC, M05BX, G03XC, A11CC, and H05AA. Furthermore, based on the average menopausal age of women in Korea (49.2 years) [28], each group was further subdivided into two subgroups based on the menopause cut-off age of 55 years. The exclusion criteria were as follows: (i) cases in which the medical institution type code was not a tertiary hospital, general hospital, hospital, long-term care hospital, clinic, Korean Medicine hospital, or Korean Medicine clinic; (ii) patients with missing values; and (iii) patients aged < 20 years.

### 2.3. Study Covariates

Patient characteristics, including age, sex, type of anticancer drug taken, payer type, and year included in the study, were used as grouping variables. Patient visits to medical institutions were classified as outpatient or inpatient and medical institutions were categorized as tertiary hospitals, general hospitals, hospitals, clinics, Korean Medicine hospitals, and Korean Medicine clinics. The total number of patients and medical costs for each group are presented as line graphs, with detailed figures summarized in the Appendix A. The medical costs comprised the combined sum of the insurance benefit (covered by the NHI service) and the copayment (paid by the patient). These medical expenses are determined by a review of the total medical expenses claimed by the medical care institution. Anticancer drugs and other drugs prescribed for breast cancer treatment were categorized according to the ATC code of each drug component. The number of prescribed patients and change in cost by year for each classification item were also examined. The drug costs are presented as separate cumulative graphs. The drug classification table, organized according to the ATC codes, is presented in Appendix A.

### 2.4. Statistical Analysis

Baseline characteristics of patients are presented as the number and ratio (a grouping variable). A Chi-square test was performed to evaluate differences between groups. For total medical use, total patients, total expenses, and per-patient expenses by the patient group were calculated and presented by year, and the compound annual growth rate (CAGR) according to the trend of increase and decrease by year was extracted to indicate changes in total patients, total expenses, and per-patient expenses. The formula for CAGR is as follows: ${\left\{\left(The\;value\;in\;2019\right)/\left(The\;value\;in\;2010\right)\right\}}^{\left(1/Number\;of\;years\right)}-1$. For drug prescriptions, the annual average total patients and annual average costs for each drug category were calculated. Since the range of annual fluctuations was not constant, the annual average change rate (ACR) was used instead of CAGR to illustrate the changes. The formula for calculating ACR is as follows: $\left[\left\{Log\left(VALUE2011\right)-log\left(VALUE2010\right)\right\}+\left\{log\left(VALUE2012\right)-log\left(VALUE2011\right)\right\}\;\cdots \;\left\{log\left(VALUE2019\right)-log\left(VALUE2018\right)\right\}\right]/9$. All costs in this study were converted to the 2020 average South Korean won to US dollar exchange rate and corrected to reflect the consumer price index in the health sector (Appendix A). Data were analyzed using the SAS software (version 9.4, SAS Institute, Cary, NC, USA).

## 3. Results

A total of 29,896 patients had more than one claim between 2010 and 2019, with breast cancer as the principal diagnosis. We excluded 11 patients with an incorrect medical institution type code, 23 patients with missing values, and 4 patients aged < 20 years. Among the finally included 29,858 patients, 8273 and 21,585 were classified into the OSP and non-OSP patient groups, respectively (Figure 1).

### 3.1. Patient Characteristics

The highest proportion of patients included in the study was observed in the age group of 45–54 years (*n* = 11,113 patients [37.22%]) (Table 1). Among these patients, the age distribution of the non-OSP patient group (age: 45–54 years; *n* = 8678 [40.20%]) was similar to that of the entire patient group. However, the OSP patient group (age: 55–64 years; *n* = 2822 [34.11%] patients) had a higher age distribution than that of the entire patient group (Table 1).

The total number of patients increased by an annual average rate of 7.65%, regardless of osteoporosis treatment (Table 1), and the annual average change rate for those aged ≥ 55 years was higher than that for those aged < 54 years (age < 55 years: 5.10%, age ≥ 55 years: 10.87%). The increase in the annual average rate in patients aged ≥ 55 years was greater than that in those aged < 55 years, regardless of the presence of osteoporosis treatment (Figure 2A). Regarding the annual average total treatment cost for each patient group, the growth rates were 5.61% and 2.96% for the non-OSP and OSP patient groups, respectively, in those aged < 55 years. In patients aged ≥ 55 years, the growth rates were 10.90% and 10.82% for the non-OSP and OSP patient groups, respectively. These findings indicate that the cost increased more significantly in the patient group aged ≥ 55 years (Figure 2B). Regarding the cost per patient, the OSP group with age < 55 years had the highest medical expenditure per patient among all patient groups. Additionally, the annual trend analysis showed that medical expenditure per patient declined in 2015 in all patient groups (Figure 2C). Detailed information on these data is presented in Appendix A. Regarding the type of patient visits, most patients who visited medical institutions for breast cancer received outpatient treatment (91.84%). Among medical institutions, the highest number of patients visited tertiary hospitals (63.36%), followed by general hospitals (24.84%). Moreover, there was no difference in the presence of osteoporosis treatment between the patient groups (Appendix A).

### 3.2. Hormone Therapy Prescriptions

Regarding the hormone therapy prescriptions, SERMs had the highest frequency of prescription (8949 [29.97%] patients were prescribed SERMs at least once in the entire patient group), followed by AIs (*n* = 6221 [20.84%]) (Table 1). The number of patients prescribed SERMs was high (*n* = 6803 [31.52%]) in the non-OSP patient group, and the AIs treatment frequency (*n* = 2836 [34.28%]) was more prominent in the OSP patient group (Table 1).

Subgroup analyses based on the average menopausal age of 55 years showed that regardless of osteoporosis treatment, the majority of patients aged < 55 years in both groups were prescribed SERMs (non-OSP: *n* = 5884 patients, 44.43%; OSP: *n* = 1331 patients, 41.70%), and more patients aged ≥ 55 years were prescribed AIs (non-OSP: *n* = 2358 patients, 28.26%; OSP: *n* = 2111 patients, 41.55%). In contrast, the prescription rates of SERMs among those aged ≥ 55 years were 11.02% and 16.04% in the non-OSP and OSP patient groups, respectively, whereas those of AIs were 7.76% and 22.71% in the non-OSP and OSP patient groups, respectively (Appendix A).

### 3.3. Prescribing Trends of Anticancer Medications

Anticancer drugs were divided into different categories according to the ATC code, and the annual average number of prescribed patients, costs, and change rates were analyzed for all patients, OSP patients, and non-OSP patients aged ≥ 55 or <55 years. SERMs were the most frequently prescribed (873 patients), followed by AIs (622 patients). However, when comparing the two anticancer drugs, the annual average prescription change rates were 8.68% and 5.9% for AIs and SERMs, respectively, with a higher increase in AIs prescriptions than in SERMs prescriptions. Additionally, SERMs was the most frequently prescribed anticancer drug, with 588 and 130 cases in the non-OSP and OSP patient groups aged < 55 years, respectively. In contrast, AIs were relatively infrequently prescribed, with 103 and 73 patients in the non-OSP and OSP patient groups aged < 55 years, respectively. For those aged ≥ 55 years, AIs were more commonly prescribed, with 236 and 211 patients in the non-OSP and OSP patient groups, respectively, while the prescription rate of SERMs was relatively low, with 92 and 62 patients in the non-OSP and OSP patient groups, respectively. Moreover, the annual average medical cost for targeted anticancer drugs was the highest in the entire patient group (USD 2,294,696) and showed the largest cost growth rate among all anticancer drugs, with an annual average increase of 16.37%. Moreover, the costs of SERMs and AIs similarly showed annual average increases of 1.80% and 4.14%, respectively (Table 2).

Next, we examined the ratio (%) of the annual average cost of each anticancer drug to the total cost of the anticancer drug (Figure 3). In the non-OSP and OSP patient groups aged < 55 years, SERMs accounted for 6.52% and 4.31% of the annual average, respectively, and AIs accounted for 5.26% and 10.97% of the annual average, respectively; in the non-OSP and OSP patient groups aged ≥ 55, SERMs accounted for 3.62% and 2.35% of the annual average, respectively, and AIs accounted for 32.86% and 21.39% of the annual average, respectively (Figure 3). After excluding the non-OSP patient group aged < 55 years, the proportion of AIs’ overall cost was high, especially in the patient group aged ≥ 55 years. Details on the annual average number of prescriptions, costs, and annual average change rates for anticancer drugs in the OSP and non-OSP patient groups based on all patients and the age of 55 years can be found in the Appendix A.

## 4. Discussion

Using national health insurance claims data from HIRA, we examined the treatment tendencies of overall and anticancer drugs, with a focus on SERMs and AIs. Among all patients with breast cancer who visited the hospital during the 10 years, the number of patients per year in all patient groups steadily increased, with the exception of some sections in 2015. Treatment costs displayed an overall upward trend. SERMs were the most commonly prescribed anticancer drugs (29.97%) in the entire patient group, followed by AIs (20.84%). Additionally, the non-OSP group had the highest SERMs prescription rate (31.52%), whereas the OSP group had a higher AIs prescription rate (34.28%). Moreover, regarding the difference in the prescription rate between the two drugs based on age and the presence of OSP, SERMs (43.90%) and AIs (33.29%) were frequently prescribed before and after the age of 55 years, respectively, regardless of the presence of OSP.

Similarly to the previous study [5], this study found that the largest proportion of breast cancer patients in Korea is found among individuals aged 45 to 55 years. Consistently, the present study showed an annual average growth rate of 7.65% in 10 years in the entire patient group (Table 1). Regarding the rate of change in the number of patients stratified by the age of 55 years, the number of breast cancer patients was higher before the age of 55 years, but the rate of increase was steeper in those aged ≥ 55 years (Figure 2, Appendix A). Considering that the risk of breast cancer increases significantly with an increase in body mass index and abdominal obesity after menopause [29] and the prevalence of obesity is increasing in all sex and age groups in Korea [30], obesity may have a particular effect on the postmenopausal increase in the breast cancer incidence in patients. The overall treatment costs also showed an increasing trend (Figure 2A,B). Notably, in 2015, the medical costs per patient declined. This finding seems to be related to the introduction of Herceptin subcutaneous injections in Korea at the end of 2014. The introduction of Herceptin subcutaneous injections might have had a significant impact on the overall cost reduction in the corresponding years due to the high preference of patients for this therapy [31,32]. Because the administration procedure is simple and the treatment time is shorter, the economic burden of Herceptin subcutaneous injections is lower than that of conventional intravenous medications (Figure 2C).

We proceeded to examine the medical costs based on age and osteoporosis-related treatment for each patient group (Figure 2). The findings showed that the total medical cost was the highest in the non-OSP patient group aged < 55 years. However, the OSP patient group aged < 55 years had the highest medical cost per patient among all patient groups. It is worth noting that younger patients with breast cancer who are premenopausal have a relatively poorer treatment prognosis than do postmenopausal patients [33,34]. Additionally, young patients with breast cancer in Korea often present with more aggressive tumor characteristics and a worse survival rate [35]. High-dose adjuvant chemotherapy is considered a more favorable treatment option for premenopausal patients with breast cancer, a high-risk group, than for postmenopausal patients [36]. Moreover, as the risk of bone disease increases in the Korean premenopausal breast cancer patient group receiving chemotherapy, related costs in premenopausal cases are expected to increase compared to those in postmenopausal cases [37].

Regarding the cost of anticancer drugs, the annual average cost of targeted anticancer drugs in the entire patient group was the highest (USD 2,294,696), and the annual average change rate was also the highest, with an increase of 16.37% (Table 2). Since targeted therapies, including examinations or treatments, often entail higher costs compared to standard treatment options [38,39], they impose a relatively heavy burden for patients with breast cancer in Korea compared to other anticancer drugs. Additionally, AIs were prescribed more frequently than SERMs (8.68% vs. 6.21%; Table 2), and on average, the cost increase of AIs was larger than that of SERMs (4.14% vs. 1.80%; Table 2). Overall, the annual average number of patients prescribed AIs and the annual average cost of AIs in the group of patients aged ≥ 55 years increased significantly (Table 2, Figure 3). This observation is likely attributable to the rising prevalence of breast cancer increased in Korean women aged ≥ 55 years over the 10-year period examined in this study.

We further analyzed the annual average number of patients prescribed for each anticancer adjuvant, and the findings showed that the number was 895 for SERMs and 622 for AIs in the overall cohort, suggesting that SERMs were more frequently used for breast cancer patients than AIs (Table 2) [9]. However, considering that prescribing AIs only after menopause is a standard treatment [12], patients aged < 55 years who accounted for 55.04% of the total breast cancer patients might have a high prescription rate of SERMs. A shift of endocrinal therapy from tamoxifen to AI was not detected in this study, contrary to a previous study; instead, a differentiated choice of medications depending on the age group was more prevalent from our findings [26].

We next investigated the total prescription rate of anticancer drugs according to osteoporosis treatment; in the non-OSP patient group, SERMs were prescribed twice as much as AIs (31.52% vs. 15.68%), while in the OSP patient group, AIs were prescribed 1.5 times more than SERMs (34.28% vs. 25.94%) (Table 1). Subgroup analyses by the reference age of 55 years showed that, regardless of osteoporosis-related treatment, SERMs were prescribed more frequently in patients aged < 55 years who might not have undergone menopause relatively, and more AIs were prescribed in those aged ≥ 55 years. In other words, SERMs prescriptions were uncommon in the entire non-OSP patient group, and AIs prescriptions were uncommon in the OSP patient group. The non-OSP patient group aged ≥ 55 years had a high prescription rate of AIs, and the OSP patient group aged < 55 years had a relatively high prescription rate of SERMs. This finding suggests that age is an essential factor in prescribing anticancer adjuvants, and breast cancer patients in Korea have mainly been prescribed SERMs before menopause and AIs thereafter [12]. In detail, the prescription rate of SERMs in the non-OSP patient group aged ≥ 55 years was almost similar to that in the OSP patient group (11.02% vs. 16.04%). However, the prescription rate of AIs in the OSP group aged ≥ 55 years was approximately 1.5 times higher than that in the corresponding non-OSP patient group (41.55% vs. 28.26%), and the prescription rate of AIs in the OSP group aged < 55 years was approximately three times higher (22.71% vs. 7.76%) than that in the corresponding non-OSP patient group. Overall, the prescription rate of AIs in the OSP patient group was higher than that of SERMs in the non-OSP patient group, suggesting a correlation between the type of anticancer adjuvants and osteoporosis (Table 2, Appendix A).

The average age of menopause is delayed in Korean women, resulting in a longer duration of estrogen exposure, which may further increase the risk of breast cancer [40,41]. We analyzed the results on SERMs and AIs, which are representative treatment drugs for breast cancer, and the findings showed that there was a tendency for prescribing SERMs and AIs, depending on the presence of osteoporosis-related treatment. Age appeared to be the primary factor influencing the choice of treatment. The non-OSP patient group had a relatively high prescription rate of SERMs compared to the OSP patient group aged < 55 years, although the difference was not statistically significant. In contrast, for those aged ≥ 55 years, the prescription rate of AIs in both groups increased predominantly on an annual basis. This finding indicates that the increased rate of AIs prescription was higher than that of SERMs. AIs are more cost-effective than SERMs in all early, advanced, and metastatic breast cancers [42]. Moreover, the prevalence of breast cancer in patients in their 50s or older is rapidly increasing, and relatively older patients show higher compliance with AIs [42,43]. Given these observations, it is important to conduct long-term follow-up studies to explore the association between AIs and osteoporosis.

### Strengths and Limitations

This cross-sectional study using national sample data was designed and conducted to examine the overall prescription trend of medications of breast cancer with a special focus on osteoporosis and hormonal medications. The strengths include the representative data used in this study which encompasses the entire population. The HIRA-NPS was used to analyze the medical use status of patients with breast cancer for a total of 10 years (2010–2019). The present data covered all groups aged ≥ 20 years, identified the overall treatment status, including therapeutic treatments and economic costs, and analyzed the current status of anticancer drug prescriptions according to the rate of related anticancer drugs taken and osteoporosis treatment in breast cancer patients in Korea.

However, this study has certain limitations. First, the HIRA-NPS data from 2010 to 2019 do not encompass items not covered by insurance or indirect costs, potentially resulting in an underestimation of the number of patients and medical costs. Second, the assessment of the overall treatment status was based on patients diagnosed with breast cancer (C50) and those with a specification, including osteoporosis (M80, M81, and M82), as either a principal or sub-diagnosis in the examination year or those who received osteoporosis treatment at least once. However, because the specification does not differentiate between therapeutic and preventive purposes and the assigned diagnosis and treatment codes, the OSP group might be overestimated. Third, this study only examined the cross-sectional trend from 2010 to 2019 using the HIRA-NPS, and future changes could not be predicted. Follow-up studies are necessary to establish specific causality and associations.

## 5. Conclusions

This study examined the overall drug prescription with a focus on anticancer drugs, treatment tendency, and costs in patients with or without osteoporosis treatment for breast cancer. The findings showed that the presence of osteoporosis-related treatment and patient age were associated with anticancer drug prescription. Furthermore, the increasing trend of breast cancer was found to be more pronounced in the group aged ≥ 55 years in Korea. Additionally, regarding prescription trends for SERMs and AIs among anticancer drugs in Korea, AIs showed a higher rate of prescription increase. The present findings may help in the clinical diagnosis and treatment of breast cancer.

## Figures and Tables

**Figure 1 medicina-59-01505-f001:**
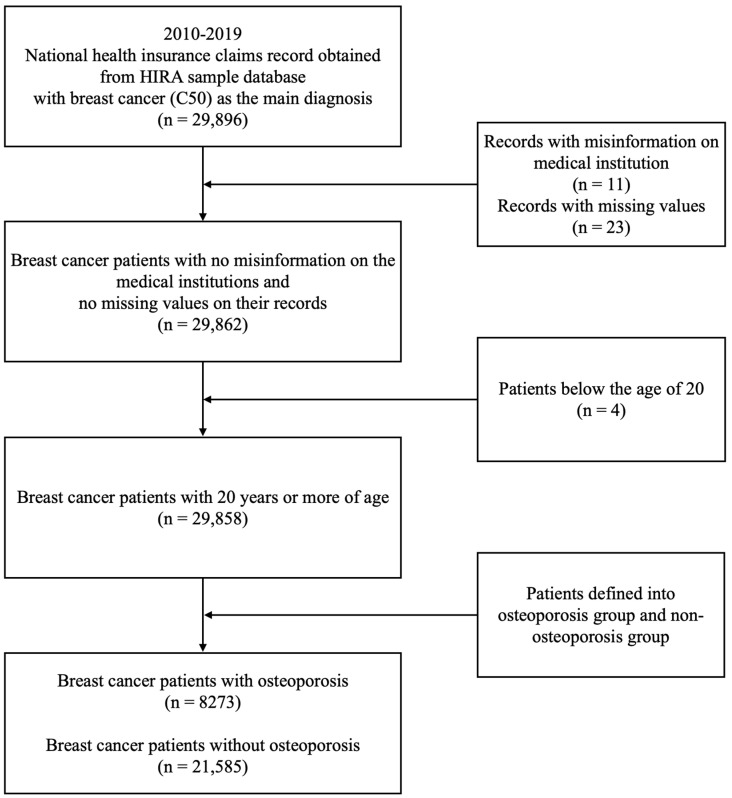
Flowchart of the study sample.

**Figure 2 medicina-59-01505-f002:**
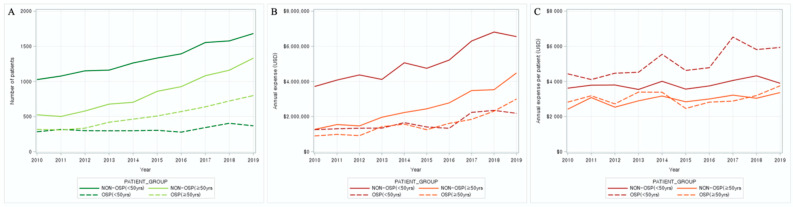
Annual medical care utilization. (**A**) Number of patients; (**B**) annual expense (USD); (**C**) annual expense per patient (USD). USD: U.S. dollar; OSP: osteoporosis.

**Figure 3 medicina-59-01505-f003:**
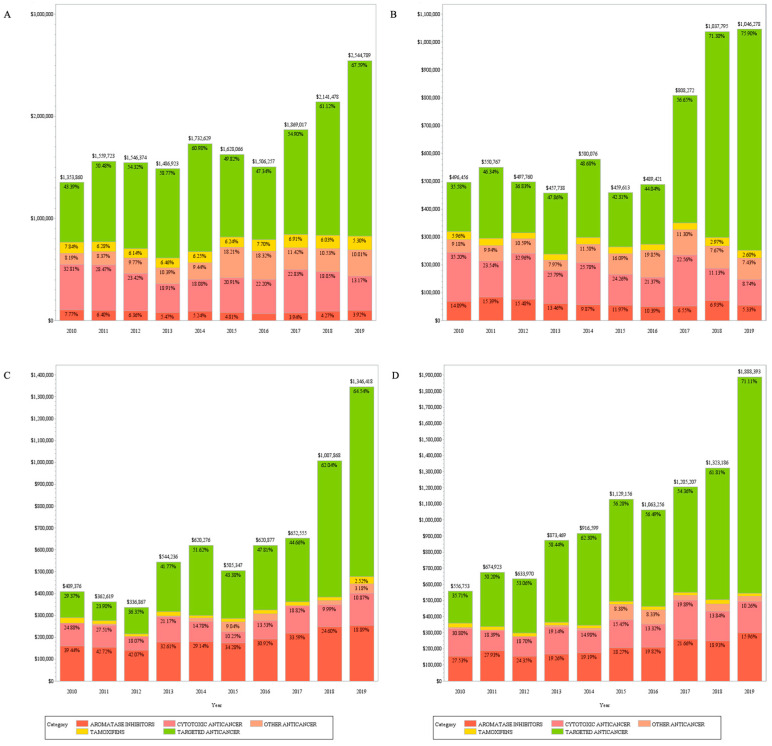
Medical cost of anticancer drugs. (**A**) Non-OSP (<55 years); (**B**) OSP (<55 years); (**C**) Non-OSP (>55 years); (**D**) OSP (>55 years).

**Table 1 medicina-59-01505-t001:** Basic characteristics of the patients.

Category	Overall Cohort		Non-OSP		OSP		*p*-Value *
	No. of Patients	%	No. of Patients	%	No. of Patients	%	
**Age (years)**							<0.0001
<35	648	2.17	563	2.61	85	1.03	
35–44	4673	15.65	4001	18.54	672	8.12	
45–54	11,113	37.22	8678	40.20	2435	29.43	
55–64	8444	28.28	5622	26.05	2822	34.11	
65–74	3719	12.46	2020	9.36	1699	20.54	
≥75	1261	4.22	701	3.25	560	6.77	
**Sex**							<0.0001
Male	120	0.40	107	0.50	13	0.16	
Female	29,738	99.60	21,478	99.50	8260	99.84	
**Anticancer**							
SERMs	8949	29.97	6803	31.52	2146	25.94	<0.0001
Aromatase Inhibitors	6221	20.84	3385	15.68	2836	34.28	<0.0001
Cytotoxic anticancer	4387	14.69	3187	14.76	1200	14.51	0.570
Targeted anticancer	1735	5.81	1211	5.61	524	6.33	0.017
Others	1502	5.03	1123	5.20	379	4.58	0.028
**Payer type**							<0.0001
NHI	28,430	95.22	20,656	95.70	7774	93.97	
Medicaid	1427	4.78	928	4.30	499	6.03	
Others	1	0.00	1	0.00	-	-	
**Year**							0.491
2010	2155	7.22	1555	7.20	600	7.25	
2011	2205	7.38	1579	7.32	626	7.57	
2012	2365	7.92	1731	8.02	634	7.66	
2013	2559	8.57	1843	8.54	716	8.65	
2014	2728	9.14	1967	9.11	761	9.20	
2015	3006	10.07	2194	10.16	812	9.82	
2016	3170	10.62	2321	10.75	849	10.26	
2017	3620	12.12	2639	12.23	981	11.86	
2018	3867	12.95	2740	12.69	1127	13.62	
2019	4183	14.01	3016	13.97	1167	14.11	
Compound annual growth rate	7.65%		7.64%		7.67%		

* Chi-square test; NHI: National health insurance. SERMs: Selective estrogen receptor modulators; OSP: osteoporosis.

**Table 2 medicina-59-01505-t002:** Annual average numbers of patients and costs of anticancer medications.

	Overall Cohort		Non-OSP (<55 Years)		OSP (<55 Years)		Non-OSP (≥55 Years)		OSP (≥55 Years)	
Anticancer Categories	No. of Patients (ACR)	Costs (ACR)	No. of Patients (ACR)	Costs (ACR)	No. of Patients (ACR)	Costs (ACR)	No. of Patients (ACR)	Costs (ACR)	No. of Patients (ACR)	Costs (ACR)
SERMs	895 (6.21%)	USD 179,295 (1.80%)	588 (6.61%)	USD 111,607 (2.66%)	133 (4.61%)	USD 25,662 (−0.80%)	92 (3.13%)	USD 20,813 (−4.09%)	62 (6.46%)	USD 17,871 (3.69%)
Aromatase Inhibitors	622 (8.68%)	USD 549,646 (4.14%)	103 (2.83%)	USD 88,607 (−0.59%)	73 (2.67%)	USD 63,709 (−2.52%)	236 (12.44%)	USD 207,035 (7.52%)	211 (9.82%)	USD 190,295 (5.05%)
Cytotoxic anticancer	439 (2.88%)	USD 775,687 (−1.40%)	219 (−0.16%)	USD 373,969 (−2.95%)	63 (0.37%)	USD 135,476 (−6.88%)	100 (7.78%)	USD 168,012 (1.89%)	57 (7.70%)	USD 97,503 (4.03%)
Other anticancer	150 (11.72%)	USD 243,773 (9.99%)	108 (12.83%)	USD 166,451 (9.21%)	35 (6.27%)	USD 59,537 (6.01%)	05 (17.88%)	USD 11,055 (69.15%)	29 (11.97%)	USD 17,107 (37.15%)
Targeted anticancer	174 (16.11%)	USD 2,294,696 (16.37%)	75 (13.41%)	USD 998,046 (11.94%)	26 (11.17%)	USD 357,198 (16.70%)	46 (21.22%)	USD 615,249 (21.23%)	27 (22.90%)	USD 317,867 (21.98%)

ACR: Annual change rate; SERMs: selective estrogen receptor modulators. The ACR was calculated by the formula: [{Log(VALUE_2011_) − log(VALUE_2010_)} + {log(VALUE_2012_)-log(VALUE_2011_)} ⋯∙{log(VALUE_2019_) − log(VALUE_2018_)}]/9. All costs were converted with the annual average exchange rate (KRW/USD, see Appendix A).

## Data Availability

The datasets analyzed in the current study are available upon authorization by the inquiry committee of research support within HIRA. The Patient Samples are provided in a DVD (text file) format, and a fee is charged for the samples. https://opendata.hira.or.kr/home.do (accessed on 10 July 2023).

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
