# Peer review of "Osteoporosis Associated with Breast Cancer Treatments Based on Types of Hormonal Therapy: A Cross-Sectional Study Using Korean National Sample Data"

_medicina, 2023, doi:10.3390/medicina59091505_

Round 1

Reviewer 1 Report

The authors of the manuscript entitled, 'Osteoporosis associated with breast cancer treatments based on types of hormonal therapy: a cross-sectional study using national sample data' have tried to investigate osteoporosis-related treatments in cancer patients with special reference to selective estrogen receptor modulators (SERMs) and aromatase inhibitors (AIs) in the Korean population. It is an important study focusing on the current problem of increasing cancer and its affiliated co-morbidities. I have some following suggestions/pointers that will make this study more clear and readable.

1.         Reference must be cited for the line starting from 'while osteoporosis is a common………………..increased risk of fractures' (lines 62-65).

2.         In Line 79, the word 'claims' must be defined, otherwise it looks incomplete.

3.         Line starting from, ‘Using claims data…….women and osteoporosis’ is incomprehensible so re-write it (line 79-83).

4.         Please clarify the meaning of M80, M81 or M82 (line 106).

5.         In the section, Study design and population' a proper flow diagram showing data collection protocol with inclusion/exclusion criteria must be given. Although the authors have given Figure 1 in the Results section, it seems incomplete (missing inclusion/exclusion criteria) and not in the proper position of the manuscript. It should be in the Study Design and Population section with edits as suggested.

6.         Line 143, (1/number of years) is above the normal line (super scripted), which needs to be corrected.

7.         Line 111-114, the authors are writing that the average menopausal age in Korean women is 49.2 years and at the same time write that 'because there is high probability of premenopause before the age of 55 years and postmenopause after 55 years'. It is losing clarity, so please rewrite this line with more precision, otherwise, women of the age 50 or 52 can be grouped as postmenopause.

8.         It would be better if the authors provide the scope and limitations of the study. This would clarify the extent to which the findings can be generalized and any potential constraints in the research methodology.

Minor English language editings are required

Author Response

Reviewer 1

The authors of the manuscript entitled, 'Osteoporosis associated with breast cancer treatments based on types of hormonal therapy: a cross-sectional study using national sample data' have tried to investigate osteoporosis-related treatments in cancer patients with special reference to selective estrogen receptor modulators (SERMs) and aromatase inhibitors (AIs) in the Korean population. It is an important study focusing on the current problem of increasing cancer and its affiliated co-morbidities. I have some following suggestions/pointers that will make this study more clear and readable.

- We deeply appreciate the reviewer’s comments.

  1. Reference must be cited for the line starting from 'while osteoporosis is a common………………..increased risk of fractures' (lines 62-65).

- We appreciate the reviewer’s comment and revised the manuscript as follows:

While osteoporosis is a common risk for postmenopausal women with or without breast cancer, its occurrence in breast cancer patients is primarily attributed to AIs since they deplete residual estrogen and are associated with rapid bone loss and increased risk of fractures[20,21].

  1. Hadji P. Aromatase inhibitor-associated bone loss in breast cancer patients is distinct from postmenopausal osteoporosis. Critical reviews in oncology/hematology. 2009;69(1):73-82.
  2. Coleman RE, Body JJ, Gralow JR, Lipton A. Bone loss in patients with breast cancer receiving aromatase inhibitors and associated treatment strategies. Cancer Treat Rev. 2008;34 Suppl 1:S31-42. Epub 20080516. doi: 10.1016/j.ctrv.2008.03.005. PubMed PMID: 18486346.

  1.         In Line 79, the word 'claims' must be defined, otherwise it looks incomplete.

- We appreciate the reviewer’s comment and revised the manuscript as follows:

Using national health insurance claims data from the Health Insurance Review and Assessment Service (HIRA), we investigated prescription status of osteoporosis-related treatments based on osteoporosis diagnosis and age.

  1. Line starting from, ‘Using claims data…….women and osteoporosis’ is incomprehensible so re-write it (line 79-83).

- We appreciate the reviewer’s comment and revised the manuscript as follows:

Using national health insurance claims data from the Health Insurance Review and Assessment Service (HIRA), we investigated prescription status of osteoporosis-related treatments based on osteoporosis diagnosis and age.

  1. Please clarify the meaning of M80, M81 or M82 (line 106).

- We appreciate the reviewer’s comment and revised the manuscript as follows:

a principal or sub-diagnosis of osteoporosis (ICD-10 code: M80 (Osteoporosis with pathological fracture), M81(Age-related osteoporosis without current pathological fracture), or M82(Osteoporosis in diseases classified elsewhere))

  1. In the section, Study design and population' a proper flow diagram showing data collection protocol with inclusion/exclusion criteria must be given. Although the authors have given Figure 1 in the Results section, it seems incomplete (missing inclusion/exclusion criteria) and not in the proper position of the manuscript. It should be in the Study Design and Population section with edits as suggested.

- We appreciate the reviewer’s comment and revised the manuscript as follows:

We excluded 11 patients with an incorrect medical institution type code, 23 patients with missing values, and 4 patients aged <20 years. Among the finally included 29,858 patients, 8,273 and 21,585 were classified into the OSP and non-OSP patient groups, respectively (Figure 1).

Figure 1. Flowchart of the study sample.

  1. Line 143, (1/number of years) is above the normal line (super scripted), which needs to be corrected.

- We appreciate the reviewer’s comment. The formula for compound annual growth rate involves, however, (the value in 2019 divided by the value of 2010) to the power of (the reciprocal of number of years) minus 1. Based on the reviewer’s comment, we inserted the formula using the equation function embedded in Words as follows:

The formula for CAGR is as follows: .

  1. Line 111-114, the authors are writing that the average menopausal age in Korean women is 49.2 years and at the same time write that 'because there is high probability of premenopause before the age of 55 years and postmenopause after 55 years'. It is losing clarity, so please rewrite this line with more precision, otherwise, women of the age 50 or 52 can be grouped as postmenopause.

- We appreciate the reviewer’s comment and revised the manuscript as follows:

Furthermore, based on the average menopausal age of women in Korea (49.2 years) [27], each group was further subdivided into two subgroups based on the menopause cut-off age of 55 years.

  1.       It would be better if the authors provide the scope and limitations of the study. This would clarify the extent to which the findings can be generalized and any potential constraints in the research methodology.

- We appreciate the reviewer’s comment and revised the manuscript as follows:

This cross-sectional study using national sample data was designed and conducted to examine the overall prescription trend of medications of breast cancer with a special focus on osteoporosis and hormonal medications. The strengths include the representative data used in this study which encompasses the entire population. The HIRA-NPS was used to analyze the medical use status of patients with breast cancer for a total of 10 years (2010–2019). The present data covered all groups aged ≥20 years, identified the overall treatment status, including therapeutic treatments and economic costs, and analyzed the current status of anticancer drug prescriptions according to the rate of related anticancer drugs taken and osteoporosis treatment in breast cancer patients in Korea.

However, this study has certain limitations. First, the HIRA-NPS data from 2010 to 2019 do not encompass items not covered by insurance or indirect costs, potentially resulting in an underestimation of the number of patients and medical costs. Second, the assessment of the overall treatment status was based on patients diagnosed with breast cancer (C50) and those with a specification, including osteoporosis (M80, M81, and M82), as either a principal or sub-diagnosis in the examination year or those who received osteoporosis treatment at least once. However, because the specification does not differentiate between therapeutic and preventive purposes and the assigned diagnosis and treatment codes, the OSP group might be overestimated. Third, this study only examined the cross-sectional trend from 2010 to 2019 using the HIRA-NPS, and future changes could not be predicted. Follow-up studies are necessary to establish specific causality and associations.

Reviewer 2 Report

The study is grounded in prior evidence and reflects a unique Korean population. Despite relatively sound measures (other than a strong rationale of the 55 year old cutoff--when the average age of 49 years old is stated for menopause in that population), there is nothing new. The study findings confirm prior research and mirror image clinical practice. Older women (i.e. postmenopausal) are more frequently prescribed AI therapy and AI therapy is associated with a higher incidence of osteoporosis and osteoporosis treatment. While unlikely available in the dataset, would be good to know how many postmenopausal women were on SERM therapy because of side effects of previously prescribed AI treatment. 

---------------------

1. What is the main question addressed by the research? Main questions was the prescription of anti-estrogen therapy in breast cancer with SERM or Ai and treatment of osteoporosis. 2. Do you consider the topic original or relevant in the field? It is an original investigation for a Korean population of women with breast cancer. Does it address a specific gap in the field? It is challenging to say that it addresses a gap in the filed. The research question and purpose might be better stated to explore compliance with national guidelines for prescription of SERMs and AI's by age group and the appropriate treatment of osteoporosis. 3. What does it add to the subject area compared with other published material? The findings really don't add any new information to what is known--younger women (premenopausal) generally are prescribed SERMs and have a low risk of osteoporosis (due to menopausal status and SERM), older postmenopausal women are generally prescribed AI's and have a higher risk of developing osteoporosis largely related to Ais and also some contribution of age-related bone loss. 4. What specific improvements should the authors consider regarding the methodology? Identify if there are more specific variables --such as any sequencing of SERMs and Ais, diagnosis of osteopenia (and any treatment) and fractures in those with diagnoses of osteoporosis. What further controls should be considered? Possibly controlling for age categories (i.e. 55-65, 65-75, > 75 years) 5. Are the conclusions consistent with the evidence and arguments presented and do they address the main question posed? yes 6. Are the references appropriate? Yes 7. Please include any additional comments on the tables and figures. None
---------------------

Author Response

  1. What is the main question addressed by the research? Main questions was the prescription of anti-estrogen therapy in breast cancer with SERM or Ai and treatment of osteoporosis. 2. Do you consider the topic original or relevant in the field? It is an original investigation for a Korean population of women with breast cancer. Does it address a specific gap in the field? It is challenging to say that it addresses a gap in the filed. The research question and purpose might be better stated to explore compliance with national guidelines for prescription of SERMs and AI's by age group and the appropriate treatment of osteoporosis. 3. What does it add to the subject area compared with other published material? The findings really don't add any new information to what is known--younger women (premenopausal) generally are prescribed SERMs and have a low risk of osteoporosis (due to menopausal status and SERM), older postmenopausal women are generally prescribed AI's and have a higher risk of developing osteoporosis largely related to Ais and also some contribution of age-related bone loss. 4. What specific improvements should the authors consider regarding the methodology? Identify if there are more specific variables --such as any sequencing of SERMs and Ais, diagnosis of osteopenia (and any treatment) and fractures in those with diagnoses of osteoporosis. What further controls should be considered? Possibly controlling for age categories (i.e. 55-65, 65-75, > 75 years) 5. Are the conclusions consistent with the evidence and arguments presented and do they address the main question posed? yes 6. Are the references appropriate? Yes 7. Please include any additional comments on the tables and figures. None

---------------------

- We appreciate the reviewer’s comment. Based on the reviewer’s comment, we further polished the overall manuscript to compare this with previous studies and add clinical implications. Furthermore, a professional editing of the manuscript was applied to refine English writing. Below are some examples of revised passages:

Introduction

However, only one study has been published 8 years ago on the prescription trend of selective estrogen receptor modulators (SERMs) and AIs, in which a shift from tamoxifen to AI was detected[26]. An insufficient number of studies on osteoporosis-related treatments in breast cancer patients and lack of recent updates deter clinicians and researchers from understanding the latest epidemiological evidence on its prevalence and relation to anticancer treatments.

Discussion

We further analyzed the annual average number of patients prescribed for each anticancer adjuvant, and the findings showed that the number was 895 for SERMs and 622 for AIs in the overall cohort, suggesting that SERMs were more frequently used for breast cancer patients than AIs (Table 2) [9]. However, considering that prescribing AIs only after menopause is a standard treatment [12], patients aged <55 years who accounted for 55.04% of the total breast cancer patients might have a high prescription rate of SERMs. A shift of endocrinal therapy from tamoxifen to AI was not detected in this study, contrary to a previous study; instead, a differentiated choice of medications depending on the age group was more prevalent from our findings[26].

This cross-sectional study using national sample data was designed and conducted to examine the overall prescription trend of medications of breast cancer with a special focus on osteoporosis and hormonal medications. The strengths include the representative data used in this study which encompasses the entire population. The HIRA-NPS was used to analyze the medical use status of patients with breast cancer for a total of 10 years (2010–2019). The present data covered all groups aged ≥20 years, identified the overall treatment status, including therapeutic treatments and economic costs, and analyzed the current status of anticancer drug prescriptions according to the rate of related anticancer drugs taken and osteoporosis treatment in breast cancer patients in Korea.

However, this study has certain limitations. First, the HIRA-NPS data from 2010 to 2019 do not encompass items not covered by insurance or indirect costs, potentially resulting in an underestimation of the number of patients and medical costs. Second, the assessment of the overall treatment status was based on patients diagnosed with breast cancer (C50) and those with a specification, including osteoporosis (M80, M81, and M82), as either a principal or sub-diagnosis in the examination year or those who received osteoporosis treatment at least once. However, because the specification does not differentiate between therapeutic and preventive purposes and the assigned diagnosis and treatment codes, the OSP group might be overestimated. Third, this study only examined the cross-sectional trend from 2010 to 2019 using the HIRA-NPS, and future changes could not be predicted. Follow-up studies are necessary to establish specific causality and associations.

Reviewer 3 Report

The present study aims to investigate osteoporosis-related treatments and the overall anticancer drug treatment tendencies, with a focus on selective estrogen receptor modulators and aromatase inhibitors, in Korean patients with breast cancer from 2010 to 2019. The work is well organized, and the manuscript is well written. Since the study exclusively uses data from the Korean population, it is reasonable to include this in the title, as well as in the limitations of the study. Otherwise, the authors should discuss their findings with other findings from similar studies in other countries.

Author Response

The present study aims to investigate osteoporosis-related treatments and the overall anticancer drug treatment tendencies, with a focus on selective estrogen receptor modulators and aromatase inhibitors, in Korean patients with breast cancer from 2010 to 2019. The work is well organized, and the manuscript is well written. Since the study exclusively uses data from the Korean population, it is reasonable to include this in the title, as well as in the limitations of the study. Otherwise, the authors should discuss their findings with other findings from similar studies in other countries.

- We appreciate the reviewer’s comment and revised the title as follows:

Osteoporosis associated with breast cancer treatments based on types of hormonal therapy:
a cross-sectional study using Korean national sample data

- Furthermore, we discussed related publication introduction and revised our manuscript based on the reviewer’s comment.

Introduction

However, only one study has been published 8 years ago on the prescription trend of selective estrogen receptor modulators (SERMs) and AIs, in which a shift from tamoxifen to AI was detected[26]. An insufficient number of studies on osteoporosis-related treatments in breast cancer patients and lack of recent updates deter clinicians and researchers from understanding the latest epidemiological evidence on its prevalence and relation to anticancer treatments.

Discussion

We further analyzed the annual average number of patients prescribed for each anticancer adjuvant, and the findings showed that the number was 895 for SERMs and 622 for AIs in the overall cohort, suggesting that SERMs were more frequently used for breast cancer patients than AIs (Table 2) [9]. However, considering that prescribing AIs only after menopause is a standard treatment [12], patients aged <55 years who accounted for 55.04% of the total breast cancer patients might have a high prescription rate of SERMs. A shift of endocrinal therapy from tamoxifen to AI was not detected in this study, contrary to a previous study; instead, a differentiated choice of medications depending on the age group was more prevalent from our findings[26].